# Data-driven discovery of changes in clinical code usage over time: a case-study on changes in cardiovascular disease recording in two English electronic health records databases (2001–2015)

Patrick Rockenschaub [ID],[1,2] Vincent Nguyen,[1,2] Robert W Aldridge [ID],[1,2] Dionisio Acosta [ID],[1,2] Juan Miguel García-Gómez,[3] Carlos Sáez[3]

For numbered affiliations see end of article.

**Correspondence to**
Patrick Rockenschaub;
patrick.rockenschaub.15@ucl.ac.uk

## ABSTRACT

**Objectives** To demonstrate how data-driven variability methods can be used to identify changes in disease recording in two English electronic health records databases between 2001 and 2015.

**Design** Repeated cross-sectional analysis that applied data-driven temporal variability methods to assess month-by-month changes in routinely collected medical data. A measure of difference between months was calculated based on joint distributions of age, gender, socioeconomic status and recorded cardiovascular diseases. Distances between months were used to identify temporal trends in data recording.

**Setting** 400 English primary care practices from the Clinical Practice Research Datalink (CPRD GOLD) and 451 hospital providers from the Hospital Episode Statistics (HES).

**Main outcomes** The proportion of patients (CPRD GOLD) and hospital admissions (HES) with a recorded cardiovascular disease (CPRD GOLD: coronary heart disease, heart failure, peripheral arterial disease, stroke; HES: International Classification of Disease codes I20-I69/G45).

**Results** Both databases showed gradual changes in cardiovascular disease recording between 2001 and 2008. The recorded prevalence of included cardiovascular diseases in CPRD GOLD increased by 47%–62%, which partially reversed after 2008. For hospital records in HES, there was a relative decrease in angina pectoris (−34.4%) and unspecified stroke (−42.3%) over the same time period, with a concomitant increase in chronic coronary heart disease (+14.3%). Multiple abrupt changes in the use of myocardial infarction codes in hospital were found in March/April 2010, 2012 and 2014, possibly linked to updates of clinical coding guidelines.

**Conclusions** Identified temporal variability could be related to potentially non-medical causes such as updated coding guidelines. These artificial changes may introduce temporal correlation among diagnoses inferred from routine data, violating the assumptions of frequently used statistical methods. Temporal variability measures

### Strengths and limitations of this study

► We are able to show previously unreported changes in coding of cardiovascular disease in two of the largest electronic health record databases in the UK.
► Temporal variability methods supports the semi-automatic identification of time trends in data recording within large and complex electronic health record databases.
► The methods can be applied to multitype and multimodal data, are robust to large sample sizes and can be performed simultaneously for multiple variables.
► Used metrics do not assess the correctness of codes entered by general practitioners, which require other validation techniques, but are able to signal the need for further validation.

provide an objective and robust technique to identify, and subsequently account for, those changes in electronic health records studies without any prior knowledge of the data collection process.

## INTRODUCTION

Routinely collected electronic health records (EHR) are increasingly used for clinical research.[1][2] They often pool data from different healthcare sites over multiple years, providing a readily available and representative national sample of clinical practice. The validity of results from observational studies heavily depends on the quality of the data,[3] and researchers have become increasingly aware of the importance of adequate data quality for obtaining reliable and reproducible findings.[4][5] Systematic approaches to ascertain data quality in health data repositories have traditionally focused on the data quality dimensions of completeness,

correctness and concordance.[6] For example, validation studies of English EHR databases commonly focused on whether all relevant information on the patient is recorded (completeness), to which degree the recorded information reflects reality (correctness) and whether the recorded information agrees with information in a reference dataset (concordance).[7] While answering these questions is of vital importance, they are not the only potential sources of bias.

Other factors that influence the reuse of data are less obvious and have often been neglected despite their potentially large impact on study results. Changes in clinical procedures over time, differences in processes between healthcare sites and the introduction of new guidelines can all cause unwarranted variations in the way data items are recorded, leading to artificial trends, irregularities and breaks in the data distributions. We have previously argued that these types of data variability over time and between participating healthcare sites pose a considerable threat to the validity and reproducibility of EHR studies.[8] Studies that ignore these variations are susceptible to obtain results of limited applicability. For example, financial incentives to improve diabetes care might exaggerate increases in recorded and reported type II diabetes.[9] At worst, variability in how data is collected might even introduce spurious relationships, such as when the above mentioned improved coding of type II diabetes reduces the incidence of patients with wrongly classified type I diabetes.[10]

In this study, we set out to demonstrate how data-driven methods can be used to identify irregularities in coding for clinical diagnoses over time using a recently developed, scalable approach that allows for easy comparison of similarities and differences in the distribution of demographic patient characteristics and cardiovascular diagnosis codes.[8 11] Using this method, we show how changes in coding guidelines can and have affected cardiovascular disease recording in two major English EHR databases in primary care (Clinical Practice Research Datalink (CPRD GOLD)[12]) and secondary care (Hospital Episode Statistics (HES)[13]) and discuss potential causes of detected variations in coding over time.

## METHODS
### Data sources
#### Clinical Practice Research Datalink (CPRD GOLD)
CPRD GOLD is a database of retrospective health records obtained directly from the practice management software (Vision, InPractice Systems LTD) of 674 primary care practices across the UK.[12] Recorded information includes patients' demography, clinical symptoms, investigations, diagnoses, and tests entered by the clinician. All clinical information is coded using Read Codes, the clinical terminology used in UK primary care until April 2018.[14] As of 2015, CPRD GOLD collected data from 674 practices, including data on 4.4 million actively contributing patients and 6.9 million historic patient records.[12]

Primary care data used in this study was taken from the subset of 400 English practices that had existing linkage to census data. All data were obtained via the CALIBER research resource.[15]

#### Hospital Episode Statistics (HES)
HES Admitted Patient Care (APC) is a repository of hospital activity data collected as part of management, planning and reimbursement of NHS hospitals in England.[13] Information is organised in finished consultant episodes (ie, the time spent under the uninterrupted care of a single consultant) and includes patients' demography, admission and discharge dates, hospital diagnoses and performed procedures. Each episode has an assigned primary diagnosis, which denotes the main condition treated during that episode, and up to 19 secondary diagnoses that contain any comorbidities relevant to the episode. Diagnoses are coded using the International Classification of Diseases 10th revision (ICD-10) codes and surgical procedures are recorded using OPCS-4 codes. In the financial year 2014/15, a total of 18.7 million episodes from 451 NHS hospital providers were captured in HES, which was equal to 34.3 episodes per 100 person-years.[13] Data from HES used in this study was preaggregated by month, age group, gender, socioeconomic status and 3-character ICD-10 code.

### Study design and population
Using the above datasets, we conducted two cross-sectional analyses of electronic health records from English primary care (CPRD GOLD) and secondary care (HES) between 2001 and 2015. The data from each database were divided into monthly cross-sectional slices (figure 1A).

For CPRD GOLD, patients contributed to any given month if they were between 20 and 110 years old at the beginning of the month and had been registered with their GP for at least 1 year. All patient data was ascertained on the first day of each month. Patients who left the practice during the month still contributed to that month. Data for each month included age (20–39, 40–59, 60–79, ≥80), gender, socioeconomic status (quintiles of the patient's Index of Multiple Deprivation (IMD) 2015[16]) and presence of coronary heart disease (CHD), heart failure, peripheral arterial disease (PAD) and stroke. Patients were required to have a complete record on age, gender and socioeconomic status, excluding those that did not (<0.1% of patients). Patients and practices were further required to fulfil standard data quality checks as performed and reported by CPRD.[12] The four cardiovascular conditions were ascertained separately and coded as present or absent. Presence of cardiovascular disease was defined as the presence of a relevant diagnosis code at any time before that month. Included diagnosis codes were directly taken from the Quality and Outcomes Framework (QOF; V.36.0), a financial incentive scheme introduced in 2004 aimed at improving the management and recording of chronic disease in primary care (see online supplementary table 1). An absence of a diagnosis code in

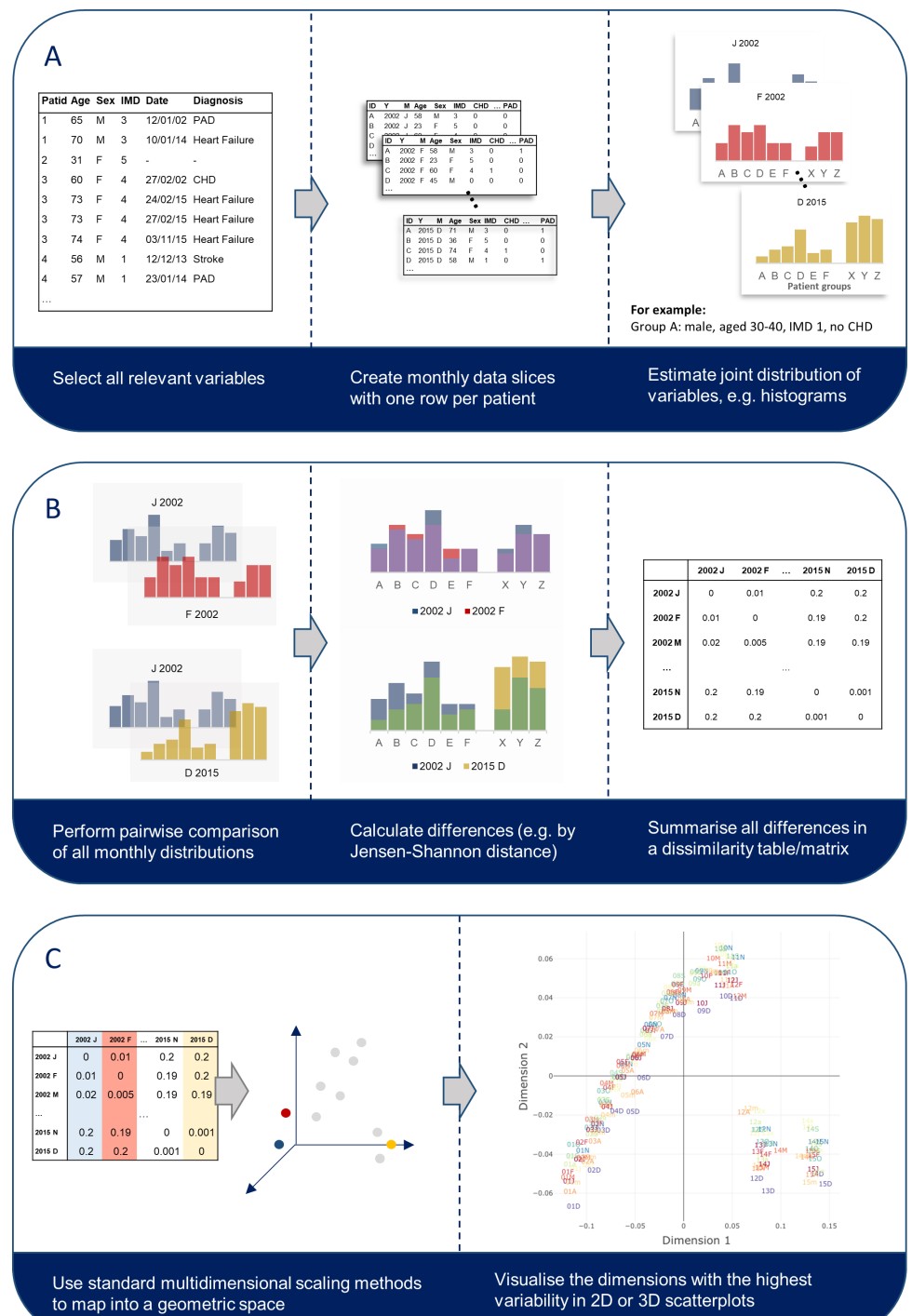

**Figure 1** Step-by-step explanation to estimate and visualise the temporal variability of a dataset. Methods included in the R package *EHRtemporalVariability* were created to support researchers with all steps in this process.

a patient's medical history was interpreted as absence of the disease.

For HES, all recorded diagnosis of ICD chapters I20-I69 (including CHD, heart failure, stroke, but not PAD; online supplementary table 2) and G45 (transient ischaemic attacks) associated with an admission of patients aged 40 years or more were counted by month and stratified by age (40–44, 45–49, 50–54, 55–59, 60–64, 65–69, 70–74,≥75), gender and socioeconomic status (quintiles of the patient's IMD 2015). The age range

and classification differed from that used in CPRD due to differences in the availability of the data. We did not distinguish between primary and secondary diagnoses and included all codes recorded during an admission. A summary of the included data for each dataset can be found in online supplementary tables 3–6.

### Temporal variability metrics

Variation in coding across months in both datasets was assessed via temporal variability methods previously

proposed and tested by two of the authors.[8 17] These metrics compare changes in the frequency of a single trait (eg, heart failure) or the joint occurrence of multiple traits (eg, male and heart failure) within a patient population over time using the monthly joint distribution of all variables of interest. In the simplest case where all variables are categorical, the joint distribution is simply the histogram of all possible value combinations (figure 1A). Temporal variability quantifies the differences in those monthly distributions based on pairwise distances between them (figure 1B). For the purpose of this study, we used the Jensen-Shannon distance (JSD), an information theoretic measure that estimates the degree of similarity between two probability distributions,[11] where 0 means equal distributions and 1 means no-overlap on the distributions. Notably, all pairwise distances are bound and independent of sample size. The dissimilarity matrix resulting from all pairwise comparisons can be mapped into a Euclidean space using multidimensional scaling (figure 1C[18]), yielding a plot that visualises the data's evolution over time and allows a graphical analysis of data recording trajectories, that is, systematic patterns in the data's evolution. A detailed description of the methods can be found in Sáez *et al* (2015)[17] and Sáez *et al* (2016).[8] Functions to perform the temporal variability analysis were implemented in the R package *EHRtemporalVariability* (https://github.com/hms-dbmi/EHRtemporalVariability).

## Statistical analysis

Empirical probability distributions of both datasets were estimated for each month by calculating the joint histogram divided by the total number of observations in each month, that is, the proportions of observations with each possible combination of variables. For CPRD GOLD, this represents the prevalence of cardiovascular diseases at the beginning of each calendar month, stratified by age group, gender and socioeconomic status. The denominator was the number of registered patients at the beginning of the month. For HES, proportions represent the relative frequency of included 3-character cardiovascular ICD-10 codes (eg, I21 Acute Myocardial Infarction), again stratified by demography. The denominator was the total number of included cardiovascular codes recorded for a hospital admission in that month.

For each dataset, the temporal variability was calculated jointly for all covariates in a given month as described above. The estimated variability was plotted in a 3D scatter plot and visually inspected for data recording trajectories. Plots were searched for gradual trends, abrupt changes, seasonality, distinct subgroups and outliers.[19] Where trends, breaks or discontinuities were observed, the same analysis was performed for each variable individually in order to isolate the source of the discovered deviation.

## RESULTS

The variables extracted from CPRD GOLD showed a gradual trend from 1 month to the next between 2001 and 2007 (figure 2 and online supplementary figures 1

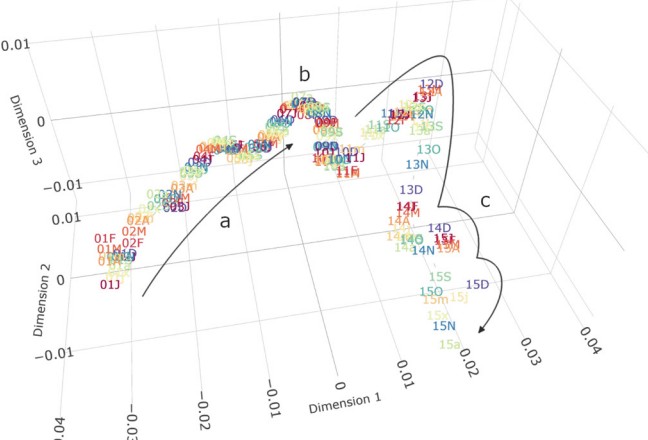

**Figure 2** IGT plot of demography (without age*) and cardiovascular disease prevalence in CPRD between 2001 and 2015. Each point represents joint prevalence in a single month (labelled with the last 2 digits of the year and the month) and distances represent the relative difference between them. Dimensions have no inherent meaning but represent the three ordered dimensions of highest variability as determined by multidimensional scaling. (a) Between 2001 and 2008, there was a gradual increase in disease prevalence, with two indentations corresponding to the years 2003 and 2005. (b) In 2008, the general trend reverses and prevalences decrease again, shown by a change in the direction of the graph. (c) The magnitude of variability increases after 2011, predominantly owing to changes in the socioeconomic status due to a reduction in the number of practices contributing to the dataset. Detailed subplots of a, b and c can be found in the supplementary material (online supplementary figure 1). CPRD GOLD, Clinical Practice Research Datalink; IGT, information-geometric temporal; J, January; F, February; M, March; A, April; m, May; j, June; x, July; a, August; S, September; O, October; N, November; D, December. *The given graph excluded the age variable for clarity. Since CPRD GOLD includes only the year of birth, including age leads to artificial yearly jumps in July when every patient is considered 1 year older. The overall conclusion remains unaltered. A full graph including age can be found in the supplementary material (online supplementary figure 2).

and 2), mostly driven by changes in prevalence of cardiovascular diseases. The pattern suggested a continuous evolution of disease prevalence compatible with social factors (eg, ageing) or incremental improvements in diagnostic coding or in clinical procedures. Smaller deviations from this overall trend could be seen at the end of 2002 and throughout 2005. Across the 8 years, the data distribution of cardiovascular disease prevalence changed with an average magnitude of about $1.5 \times 10^{-3}_{\mathrm{JSD/month}}$, which can be roughly viewed as a ~0.15% difference between consecutive histograms of disease prevalence. For comparison, if we analysed only a single variable, a change in JSD/month of this size would be obtained if its prevalence rose from 1% to 2.2%. Note further that the estimated JSD depends on base prevalence and is larger for very small or very large prevalence. An absolutely

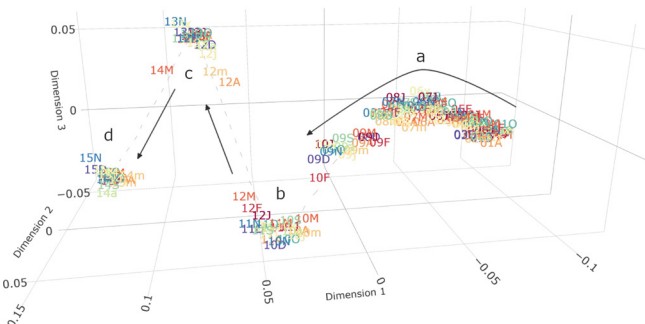

**Figure 3** IGT plot of demography and cardiovascular disease coding in HES between 2001 and 2015. Each point represents joint prevalence in a single month (labelled with the last 2 digits of the year and the month) and distances represent the relative difference between them. Dimensions have no inherent meaning but represent the three dimensions of highest variability (in order) as determined by multidimensional scaling. (a) From 2001 to 2009, there was a gradual change in which cardiovascular codes were associated with hospital admission. The data distributions started to diverge from the previous trend in March 2009. (b) In March 2010, the distribution of cardiovascular codes abruptly changed. (c and d) Similar and even stronger changes in cardiovascular disease coding occurred again in April 2012 and April 2014. The distributions within these 2 year batches remained stable. Detailed subplots of a, b, c and d can be found in the supplementary material (online supplementary figure 4). IGT, information-geometric temporal; J, January; F, February; M, March; A, April; m, May; j, June; x, July; a, August; S, September; O, October; N, November; D, December.

larger, but relatively smaller, rise in prevalence from 51% of patients to 57% would also give the same JSD/month. Changes across this period were mainly attributable to an increase in the number of patients with heart failure (from 6.7/1000 patients at the start of 2001 to 10.8/1000 by the end of 2007; +62%), stroke (from 14.4/1000 to 23.4/1000; +62%), PAD (from 7.0/1000 to 10.3/1000; +47%) and to a lesser extent CHD (from 44.7/1000 to 48.2/1000; +7.8%).

From 2008 onwards, the trend shifted direction, owing to a reduction and partial reversal in the prevalence of heart failure, PAD and most notably CHD (from 49/1000 at the beginning of 2008 to 39/1000 at the end of 2015; 20% reduction). The estimated average magnitude of change increased to $2.4 \times 10^{-3}_{\text{JSD/month}}$ between 2008 and 2015. Starting in March 2011, the gradual pattern diverged from the relatively straight path seen before due to shifts in the socioeconomic distribution of the patient population. This coincided with a substantial drop of contributing practices from more than 343 practices in January 2011 to 165 active practices in December 2015 (online supplementary figure 3).

The distribution of cardiovascular diagnoses associated with HES admissions experienced a gradual change similar to that observed for CPRD GOLD until the end of 2008 (figure 3 and online supplementary figure 4). In this period, the use of codes generally stayed comparable and

shifted only over the course of multiple years. Notable changes were seen for I20 Angina pectoris (−34.4%), I64 Unspecified stroke (−42.3%) and I25 Chronic CHD (+14.3%). After a transition period in 2009, distributions stopped to evolve gradually and started to cluster tightly by NHS financial year (April to March—figure 3). There was a major shift in cardiovascular admission coding every 2 years (financial years 2010/2011, 2012/2013 and 2014/2015). The abrupt changes were primarily due to differences in the ICD-10 codes used, while age, gender and socioeconomic status remained largely stable.

In particular, chapters I20–I25 experienced noticeable temporal breaks (figure 4). While I21 Acute myocardial infarction declined in relative frequency starting in 2006, it increased from 8.0% of included codes in March 2012 to 10.9% in April 2012 (+36%) and remained thereafter. Simultaneously, the related code I22 Subsequent myocardial infarction (including reinfarction and recurrent infarction) dropped from 1.4% in March to 0.5% in April (−64%) and finally to ~0.1% after September 2012. I20 Angina pectoris decreased from 18.8% in January 2001 to 10.2% in March 2014 (−45.7%), after which it further declined by 2% points to 8.4% in April 2014 (−17.6%).

## DISCUSSION
We discovered both gradual and abrupt changes in the distribution of cardiovascular patient populations in two large English EHR databases between 2001 and 2015 using recently developed data quality measures (table 1). The observed differences in cardiovascular disease coding might bias clinical phenotypes when applied over the entire study period, introducing correlation within time periods that violate the assumptions underlying common statistical methods (eg, regression analysis). Temporal variability measures provided an objective and robust way to identify those changes without any prior knowledge of the data.

Many studies have used CPRD GOLD to look at the incidence,[20–23] prevalence[20 21 23 24] and outcomes of cardiovascular disease.[25 26] However, none of these studies mentioned changes in the coding of diseases and only one of these studies reported findings per year.[23] Changes in coding over time tend to be underreported in research papers, their identification limited to dedicated validation studies, which depending on the disease investigated may or may not exist. To the best of our knowledge, no patterns in the recording of included cardiovascular diseases have been reported for CPRD GOLD yet. Previous validation studies in other chronic diseases did report changes in coding over time. Among others, improved coding has been reported for chronic obstructive pulmonary disease,[27] diabetes,[10] body mass index[28] and smoking status.[29] Improvements in primary care coding were primarily linked to the introduction of QOF in 2004, a payment scheme which incentivised better coding and management of chronic diseases. In line with

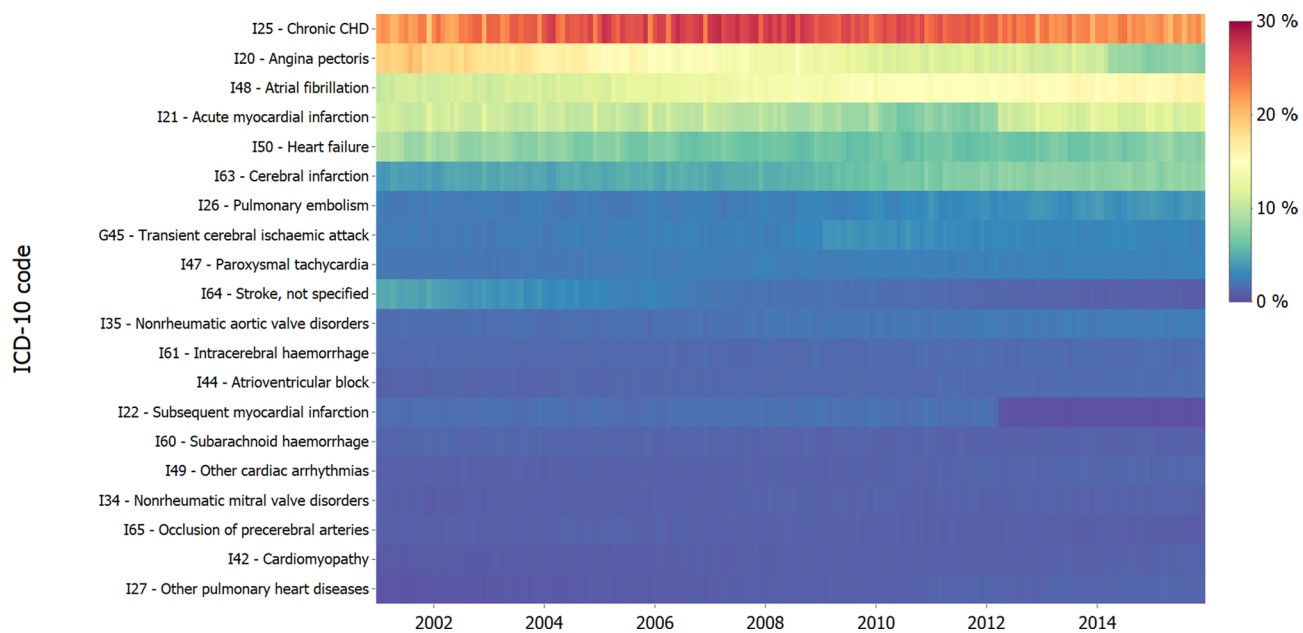

**Figure 4** DTM of ICD-10 coding linked to hospital admissions in HES between 2001 and 2015. Each row represents a single ICD-10 code (3 characters) and the colour shows the proportion of admissions with that code in each month. Gradual changes in code frequency can notably be seen for I20—angina pectoris, I21—acute myocardial infarction, I63—cerebral infarction and I64—stroke, not specified. Abrupt changes appear in the coding of G45—transient Cerebral ischaemic attack (2009), I21—acute myocardial Infarction (2010 and 2012) and I20—angina pectoris (2014). DTM, data temporal map; HES, Hospital Episode Statistics; ICD-10, International Classification of Diseases 10th revision.

these studies, we found alterations of the overall trend leading up to QOF (December 2002–December 2003) and following its introduction (October 2004–September 2005). However, these effects were small and the estimated prevalence had already been rising before 2003 and kept increasing after 2005. A study analysing the coding of diabetes further found a sharp increase of coded type 2 diabetes in 2004 that slowly started to decrease again after 2008,[9] around the same time that we observed a reversal of cardiovascular prevalence. It is unclear what prompted these changes and whether they might be related. Our method further detected notable changes in patient population after 2011, mainly due to changes in the distribution of socioeconomic status. This was likely related to a considerable reduction in the number of participating practices from around 350 to 165 at the end of the study period, potentially due to practices switching to new practice management software incompatible with CPRD GOLD.[30]

Data on cardiovascular admissions from HES have been used alone or linked with primary care data from CPRD GOLD.[24] Again, to the best of our knowledge, no changes in cardiovascular disease coding in HES have been published previously. A systematic review of discharge coding accuracy reported improved accuracy over time.[31] A validation study comparing coronary heart disease in HES (ICD-10 codes I20-I25) to data from the prospective UK Whitehall II cohort study[32] found a generally good agreement between the two data sources,[33] suggesting reasonable recording quality in HES. An earlier study comparing data from HES and the Million Women Study[34]

came to a similar conclusion.[35] However, these studies only included data up to 2013 and 2005, respectively, and did not compare data from different time periods. Other studies based on local audits or comparison to specific disease registries[36] have reported a notable underreporting of myocardial infarction cases in HES.[21 37] Any increases or decreases in the observed number of cardiovascular diseases could thus be due to improvements or deteriorations in coding. The gradual change in distributions observed until 2008 agreed with the trends observed in CPRD GOLD and could relate to substitution of vague codes with more specific codes, better diagnostics (eg, wider availability of CT scans) or a slow shift in the characteristics of the underlying patient populations. The first major change occurred in March 2010, following a new version of NHS Digital/HSCIC Coding Clinic Guidance in February.[38] This guideline included provisions for stricter coding of I21 Acute Myocardial Infarction, requiring a different coding of myocardial infarction in subsequent trusts to avoid overcounting. The additional changes in 2012 and 2014 both happened in April, coinciding with the financial year of the National Health Service and the publishing of updates to the National Clinical Coding standards,[39] making it likely that they too are the result of changes in coding practice. As these changes mostly occurred within the group I20–I25, they might not affect studies that use all of these codes, but may lead to problems if authors include only a single code from this group (eg, I21). Related preliminary results on hospital admissions for all ICD-10 codes (not only chapter I) using a traditional interrupted time series analysis showed

**Table 1** Variability in CPRD GOLD and HES and their potential causes and solutions

| Finding | Observable cause | Possible original cause | Possible solutions |
|---|---|---|---|
| *CPRD GOLD* | | | |
| Gradual change in the population distribution between 2001 and 2007 | Increases in the prevalence of recorded cardiovascular disease | Demographic changes (eg, ageing); incremental improvements in diagnostic coding or in clinical procedures | Incremental learning of models; inclusion of time interaction effect |
| Shift in the direction of change in 2008 | After the previous year's increase, the prevalence of CHD, heart failure and PAD started decreasing again around the same time | No immediate reason identified | Separate analyses of prechange and postchange data |
| Oscillations in the data distributions after 2010 | Changes in the distribution of socioeconomic status in the target distribution | Selective dropout of practices, possibly related to a switch in the practice management software | Mixed models with practice effects |
| *HES* | | | |
| Gradual change in the population distribution between 2001 and 2008 | Increase in reported chronic CHD and atrial fibrillation; decreases in reported angina pectoris, acute myocardial infarction, heart failure and stroke | Demographic changes (eg, ageing); incremental improvements in diagnostic coding or in clinical procedures; selective increase of disease incidence | Incremental learning of models; inclusion of continuous time interaction effect |
| Shift in the direction of change in 2009 | Increased coding of transient cerebral ischaemic attacks between 2009 and 2010 | No immediate reason identified | Separate analyses of prechange and postchange data |
| Abrupt change in March 2010 | Drop in acute myocardial infarction coding | Update to the HSCIC Coding Clinic Guidance in February 2010 | Separate analyses; incremental learning of models |
| Further abrupt changes in April 2012 and 2014 | Sudden increase in acute myocardial infarction coding in 2012 with concomitant drop in subsequent myocardial infarction records; sudden further decrease in angina pectoris codes in 2014 | Update to the National Clinical Coding Guidance National Clinical Coding Standards ICD-10 fourth Edition | Separate analyses; incremental learning of models |

CHD, coronary heart disease; CPRD GOLD, Clinical Practice Research Datalink; HES, Hospital Episode Statistics; ICD-10, International Classification of Diseases 10th revision; PAD, peripheral arterial disease.

further, similar changes in non-cardiovascular chapters; these results will be disseminated in a further study evaluating life-style related diseases.

It is challenging to disentangle changes solely due to how diseases are recorded from other, genuine shifts in the patient population. While abrupt changes like the one observed here for myocardial infarction strongly suggest an exogenous cause such as new clinical coding guidelines, continuous, gradual changes over a long time period can be more difficult to classify. However, we believe that it is important that researchers are aware of potential variations irrespective of the cause. Even in cases where changes are attributable to demographic shifts, accounting for them in the statistical analysis might still be warranted. The impact of observed changes, genuine as well as artificial, always depends on the specific research question.[40] For example, an increase in the estimated population prevalence of heart failure from 0.7% to 1.0% might not impact findings when accounting for heart failure as a covariate in a larger cohort, but might significantly alter the patient characteristics in a smaller, heart failure-only cohort. Similarly, while coding changes may greatly affect studies of incidence overtime, they might not change the results of a study looking at the effect of a risk factor on an outcome unless coding changes are biased towards certain patient populations. Insights gained from temporal variability analysis can be used to investigate and account for changes in patient cohorts across years.

Although some of the findings presented in this study could be detected with conventional methods such as traditional time series analysis of incidence rates, these methods usually require a formal definition of the time point at which changes happen. They further do not handle multivariate, multitype and multimodal data well[17] and require a separate analysis for each variable. This is particularly problematic when analysing changes in multinomial variables such as ICD-10 codes. More traditional methods might further struggle with large sample sizes, whereas the structure of the variability metrics allows for a flexible modelling and subsequent hypothesis testing via statistical process control. Temporal variability metrics

together with the tools provided in our *EHRtemporalVariability* R package facilitate the calculus from raw data tables directly to visualisation. Results can be shared on the Shiny user interface (http://ehrtemporalvariability.upv.es), aiding transparency and communication.

## Limitations

Results in this study are limited by the fact that the conditions chosen for inclusion represented a convenience sample based on the overlap between the two projects for which the data was originally obtained. The results shown here therefore do not constitute a systematic, in-depth validation study of cardiovascular disease recording. Indeed, the aim of this study was not to comprehensively investigate the data quality of cardiovascular coding in CPRD GOLD and HES but to show how systematic, data-driven methods for studying temporal variability can help to identify potential coding inconsistencies over time early on in a project and allow researchers to a priori adjust the analysis accordingly. We believe that routine checks of the temporal variability of study data will aid the validity and reproducibility of medical and epidemiological studies. Reporting coding variability in online supplementary material can help readers judge the reliability of codelists and strengthen the conclusions. The analysis presented here was also limited to a manual inspection of the plots, as would be appropriate for interactive data quality analysis at the start of a project. The framework can easily be extended to allow for a more formal statistical process control (see Sáez *et al* (2015 and 2018)[17 19] for guidance). Finally, changes in both databases were analysed in isolation. It is possible that the impact of some changes is mitigated if records from both databases are used jointly to define the presence or absence of disease.[21]

Despite the promising results, we must note that variability metrics are solely based on recorded data and will not detect the same data quality issues and trends identified by other, dedicated validation studies based on manual code review or GP questionnaires.[7] They are not meant to replace in-depth validation of data sources but rather complement them. Extensive validation studies are costly and are dependent on the exact codelists used during validation. Our data-driven approach might be well suited as a first step to signal the need for an extensive validation. With regard to the findings in this study, a reasonable first step in assessing their impact for a specific research study might be to perform analysis stratified by NHS financial year or observed stable periods. How results should be reported and whether the data are fit for purpose then depends on the results of these sensitivity analyses and the exact research questions investigated.

## CONCLUSION

We identified previously unreported variability in the frequency of cardiovascular codes in CPRD GOLD and HES between 2001 and 2015 using temporal variability measures that require minimal prior specification. In doing so, we have demonstrated the utility of application of data-driven approaches to data quality on two of the most important data resources for clinical research in the UK. We demonstrated that the methods can be implemented in an unsupervised, scalable manner, providing non-parametric visualisations of data recording trajectories to measure their variability. The results from this variability analysis enable researchers to adjust their analysis and ensure reproducible results.

**Author affiliations**
[1]Institute of Health Informatics, University College London, London, UK
[2]Health Data Research UK, London, UK
[3]Instituto de Aplicaciones de las Tecnologías de la Información y de las Comunicaciones Avanzadas (ITACA), Universitat Politècnica de València, Valencia, Spain

**Acknowledgements** We thank Dr Anoop Shah for his invaluable comments on reasons for the observed coding changes and the manuscript draft. This study is based on data from the Clinical Practice Research Datalink (CPRD GOLD) obtained under license from the UK Medicines and Healthcare products Regulatory Agency and on aggregated data from the Hospital Episode Statistics (HES) obtained via Public Health England. The data are provided by patients and collected by the NHS as part of their care and support. All rights reserved. All primary care data was accessed through CALIBER, which is a research resource consisting of linked electronic health record phenotypes, methods and tools, specialised infrastructure and training and support led from the UCL Institute of Health Informatics.

**Contributors** PR, DA, JMGG and CS designed the study. PR and VN performed all statistical analyses reported in this study. PR wrote the manuscript. PR, VN, RWA, DA, JMGG and CS all interpreted the data and substantially reviewed the draft manuscript. All authors read and approved the final manuscript.

**Funding** VN is funded by a Public Health England PhD Studentship. RWA is supported by a Wellcome Trust Clinical Research Career Development Fellowship (206602/Z/17/Z). JMGG and CS contributions to this work were partially supported by the MTS4up Spanish project (National Plan for Scientific and Technical Research and Innovation 2013-2016, No. DPI2016-80054-R), the CrowdHealth H2020-SC1-2016-CNECT project (No. 727560) (JMGG) and the Inadvance H2020-SC1-BHC-2018-2020 project (No. 825750). PR and DA did not receive any direct funding for this project. Access to the Clinical Practice Research Datalink was supported by the UK Economic and Social Research Council (ES/P008321/1). Access to aggregated Hospital Episode Statistics was provided by Public Health England. This work was further supported by Health Data Research UK, which is funded by the UK Medical Research Council, Engineering and Physical Sciences Research Council, Economic and Social Research Council, Department of Health and Social Care (England), Chief Scientist Office of the Scottish Government Health and Social Care Directorates, Health and Social Care Research and Development Division (Welsh Government), Public Health Agency (Northern Ireland), British Heart Foundation and the Wellcome Trust.

**Competing interests** None declared.

**Patient consent for publication** Not required.

**Ethics approval** The primary care analysis reported in this study was approved by the Medicines and Healthcare products Regulatory Agency's independent scientific advisory committee (ISAC-Nr.: 17 048), under Section 251 (NHS Social Care Act 2006). For analysis of secondary care data, the authors had only access to the aggregated data which was provided by Public Health England as part of VN's PhD, which they funded. No separate ethical approval was required.

**Provenance and peer review** Not commissioned; externally peer reviewed.

**Data availability statement** Data were obtained from a third party and are not publicly available. CPRD GOLD and HES data cannot be directly shared by the researchers but are available directly from CPRD GOLD and NHS Digital subject to standard conditions. All statistical code is available from https://github.com/prockenschaub/CPRD_HES_variability.

## ORCID iDs

Patrick Rockenschaub http://orcid.org/0000-0002-6499-7933
Robert W Aldridge http://orcid.org/0000-0003-0542-0816
Dionisio Acosta http://orcid.org/0000-0003-2817-0178

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
