## [Reviewer comments · BMJ Open]

ARTICLE DETAILS

TITLE (PROVISIONAL)	Data-driven discovery of changes in clinical code usage over time: a case-study on changes in cardiovascular disease recording in two English electronic health records databases (2001-2015)
AUTHORS	Rockenschaub, Patrick; Nguyen, Vincent; Aldridge, Robert; Acosta, Dionisio; García Gómez, Juan Miguel; Sáez, Carlos

VERSION 1 – REVIEW

REVIEWER	Dr Salwa Zghebi University of Manchester, UK
REVIEW RETURNED	05-Nov-2019

GENERAL COMMENTS	Comments for the Authors The paper presents a well-designed study looking at temporal changes in CV clinical coding in electronic health records with potential originality as this is underreported current literature. Comments • Abstract: the 'Main outcomes' section includes coronary heart disease (CHD) but later in the 'Results' section angina and MI are listed individually. Did the authors examine MI and CHD (that would include MI and angina by definition) separately? If yes, how this is justified.• Study population age criteria: could the authors explain why the age criteria are different from the CPRD GOLD and HES datasets? 20-110 years and ≥ 40 years, respectively.• On page 4, Line 54, the authors state "Primary care data used in this study was taken from the subset of 400 English practices that had existing linkage to census and hospital data." And on page 5, Line 15 "Data from HES used in this study was pre-aggregated by month, age group, gender, socio-economic status and 3-character ICD-10 code". Does this mean that the authors were not able to identify whether they included the same patients in both datasets (using patient ID and HES ID variables)? This is important as there is a risk of duplication of recorded conditions if included same patients (which is very likely) as the CV codes were captured and reported separately from both datasets. On the other hand, what are the expected effects of not taking the possibility of including same patients into account on the reported code changes?• For the CPRD GOLD analyses, the denominator was the number of patients registered at the beginning of the month. Did the patients need to be contributing to the data the whole month to be eligible for inclusion or contributing for any time e.g. for 1 day?• The numerator for HES data was the proportion of codes or CV condition? Also, shouldn't the denominator be all APC admissions recorded that month instead of the total number of included CV APC admissions in that month? Please clarify.
---

	 • Have the authors validated their EHRtemporalVariability R package they developed? For instance, was it used in previous pilot studies to test it? • Table 1 provides useful information on possible causes of observed coding variability in CPRD GOLD and HES datasets. It would be useful if the authors would give more specific steps to researchers using electronic health records to follow in future studies to take the reported variability on clinical coding into account. • It is understandable that the IGT plots contain many information and the authors have presented two formats (with and without age) and added arrows to improve visualisation. But still it is hard to spot changes at individual months of interest. If practical, I suggest considering to include a plot or two either: over a shorter period of time that showed more clinically relevant changes on coding variability (e.g. before and after QOF launch); or to plot using yearly than monthly points to improve plot clarity. • The authors stated a limitation that the CV conditions chosen for inclusion in this study represented a convenience sample from a previous project and hence no other clinical conditions were examined. Based on that, what other conditions would the authors recommend to be examined for temporal code changes in future research?
--	--

REVIEWER	Paul Aylin Imperial College London UK I am co-director of the Dr Foster Unit at Imperial, and our unit receives a proportion of its funding from Dr Foster Intelligence (a wholly owned subsidiary of Telstra Health).
REVIEW RETURNED	08-Dec-2019

GENERAL COMMENTS	The authors set out to demonstrate how data-driven variability methods can be used to identify changes in disease recording CPRD and HES between 2001-2015. They found that both databases showed gradual changes in cardiovascular disease recording between 2001 and 2008. Before commentating on the analysis, I would seriously question the number of 451 english hospital trusts found in 2014/15. There were around 180 acute trusts in 2000, now down to about 135 acute non-specialist trusts in 2018, 17 acute specialist trusts and 54 mental health trusts. Even accounting for mergers and the inclusion of mental health trusts, it seems hard to see where the figure of 451 trusts comes from. This would cast doubt on the whole analysis. Results The authors state that variables extracted from CPRD GOLD showed a gradual trend from one month to the next between 2001 and 2007. They suggest a monthly increase in the prevalence of a single disease from 1% to 2.2% or from 50% to 55%. I may be misunderstanding their statement, but a monthly increase of 55% would seem to be impossible. I suspect they mean over the whole period. What the authors show is a change in coding, particularly within HES from vague codes to more specific codes, suggesting improved coding and probably better diagnostics (with an increase in CT scans for stroke and troponin trusts in suspected AMI). Within CPRD coding appears to have been driven by QOF payments. None of this
---

	is surprising or new findings. Much earlier research has suggested an improvement in coding over time (E.M. Burns, E. Rigby, R. Mamidanna, A. Bottle, P. Aylin, P. Ziprin, O.D. Faiz. Systematic review of discharge coding accuracy. Journal of Public Health, Volume 34, Issue 1, March 2012). Although these national trends are not new findings, the methods suggested might be useful in examining temporal trends in individual trust coding practices, and therefore could be employed to help interpret trust level indicators dependent on diagnostic coding. However, I am struggling to see how the authors' suggested methods are any better than simple time series analyses, but am happy to be persuaded otherwise. I note that none of the authors are clinicians, and the paper might benefit from clinical input to help explain some of the changes,
--	--

VERSION 1 – AUTHOR RESPONSE

Comments from reviewer #1

Comment #1.1: Abstract: the 'Main outcomes' section includes coronary heart disease (CHD) but later in the 'Results' section angina and MI are listed individually. Did the authors examine MI and CHD (that would include MI and angina by definition) separately? If yes, how this is justified.

Author response:

We analysed CHD as a single category in the Clinical Practice Research Datalink (CPRD) and as subcategories (angina, acute MI, ...) in the Hospital Episode Statistics (HES). We have added a sentence to the methods section to clarify this categorisation (lines 165-166). The reason for this distinction within the two databases was as follows:

The focus of this study was to highlight changes in cardiovascular disease coding in two prominent English healthcare databases (CPRD and HES), and to show how temporal stability metrics provide an easy and powerful tool to identify said changes. Changes in cardiovascular disease coding can be analysed at different disease levels and groupings. Which level or grouping is the most appropriate depends on both the database and the research question.

For example in CPRD, coding quality is influenced by the Quality and Outcomes Framework (QOF), which includes coronary heart disease (CHD) as one major category. CHD (irrespective of the sub-category, such as angina or acute MI) in our own research has often been used as a covariate. In such a setting, it is arguably most relevant if coding of all CHD changed over time to identify any temporal breaks in the relationship between the covariate (CHD) and the exposure and/or outcome (e.g. antibiotic prescribing <https://academic.oup.com/jac/article/72/6/1818/3040192>). Consequently, the analysis of CPRD included in this study considered CHD as a binary disease and analysed if the proportion of patients with/without CHD changed over time.

When identifying acute MI within HES, researchers must rely on ICD-10 codes in chapter I20-25. When doing so, it is not immediately clear whether to use only code I21 – Acute myocardial infarction, or whether to also include I22 – Subsequent myocardial infarction, or whether to include the entire chapter I20-25 (which also includes angina, MI complications and chronic CHD; <https://icd.who.int/browse10/2016/en#/I20-I25>). In this scenario, we are more interested in the sub-categories than in the overall prevalence of CHD. Indeed, our results show that it would make a difference whether acute is defined by both I21/22 or by I21 alone, as there is a distinct temporal shift in 2012 away from I22 and towards I21.

The levels of analysis used in this study were based on commonly investigated research questions in each database, and to show how the methods employed fit all those scenarios. We were able to use different definitions because we analysed both databases independently (see the response to #1.3 for further explanation). We hope that our chosen scenarios provide helpful examples and findings, but they are not meant to give an exhaustive analysis of cardiovascular disease coding. Indeed, it would be hard to define a universally applicable definition. Depending on the research questions, analysis on other levels could be equally relevant, such as the analysis of changes in HES in the use of ICD-10 groupings (I20-25, I26-28, ...) or 4-character ICD-10 codes (I20.1, I20.2, ...). We strongly encourage future researchers to use temporal variability metrics to analyse the changes most relevant to their own research question.

Comment #1.2: Study population age criteria: could the authors explain why the age criteria are different from the CPRD GOLD and HES datasets? 20-110 years and ≥ 40 years, respectively.

Author response:

Age limits and groupings were chosen based on the data available to the researchers. We thank the reviewer for pointing out that this isn't stated clearly enough - we have added a sentence to the methods section (lines 178-179) to clarify this in the manuscript.

Comment #1.3: On page 4, Line 54, the authors state "*Primary care data used in this study was taken from the subset of 400 English practices that had existing linkage to census and hospital data.*" And on page 5, Line 15 "*Data from HES used in this study was pre-aggregated by month, age group, gender, socio-economic status and 3-character ICD-10 code*". Does this mean that the authors were not able to identify whether they included the same patients in both datasets (using patient ID and HES ID variables)? This is important as there is a risk of duplication of recorded conditions if included same patients (which is very likely) as the CV codes were captured and reported separately from both datasets. On the other hand, what are the expected effects of not taking the possibility of including same patients into account on the reported code changes?

Author response:

The two databases were analysed independently. We were therefore not able to identify the overlap between the two datasets. Given the national scope of HES, there will that there is substantial overlap in the two databases (all patients in CPRD that visited a hospital will have also contributed to HES). We do not believe, however, that this affects the validity of our results. CRPD and HES are distinct databases and are often used in isolation. Patients are only duplicate in the sense that they are present in both, the coding for these patients is kept separate (e.g. a patient could have a record of CHD in CPRD, HES, or both). Consequently, coding of a patient's cardiovascular disease can change in one or both databases.

To aid independent analysis of those databases, it is important to first establish how coding within each of the database changes irrespective of changes in the other. This is what we are able to show in this study. We can further speculate that trends observed in both datasets (e.g. the gradual change in coding between 2001 and 2008) were due to common causes such as a national programme for improving clinical coding.

We agree with the reviewer that the nature and impact of observed changes might be different if data from both sources is included in a combined analysis. We do believe, however, that many of our findings would still hold. This belief is supported by previous research, which has shown that many MI

patients, even though they were registered in both databases, were only identified in one of the two (<https://www.ncbi.nlm.nih.gov/pmc/articles/PMC3898411/>).

One expected effect of analysing both databases in isolation is that we were not able to investigate if the estimated prevalence of cardiovascular disease changed in a cohort based on both CPRD and HES records. For example, if recording (and/or diagnosis) of disease simply shifted from HES (secondary care) to CPRD (primary care), the overall estimated prevalence might remain stable but recording in either database will change. We added two sentences to the limitations to discuss this issue (lines 379-381)

Comment #1.4: For the CPRD GOLD analyses, the denominator was the number of patients registered at the beginning of the month. Did the patients need to be contributing to the data the whole month to be eligible for inclusion or contributing for any time e.g. for 1 day?

Author response:

Patient data was ascertained on the first day of each month and contributing patients were allowed to leave the practice at any time after the first day of the month. We added two sentences to clarify this in the manuscript (lines 158-159)

Comment #1.5: The numerator for HES data was the proportion of codes or CV condition? Also, shouldn't the denominator be all APC admissions recorded that month instead of the total number of included CV APC admissions in that month? Please clarify.

Author response:

The numerator for the HES data was the total number of cardiovascular disease codes associated with an admission in HES APC. This corresponds to an analysis of changes in the relative frequencies of cardiovascular disease codes, compared to all other cardiovascular disease codes.

We chose this numerator to highlight changes in the use of individual cardiovascular codes, e.g. the observed substitution of I22– Subsequent myocardial infarction by I21 – Acute myocardial infarction. While we agree with the reviewer that using all APC admissions as the denominator could lead to different (equally valid) conclusions, we believe using only cardiovascular codes as the denominator to be the most informative for our chosen scenario, as it is not influenced by changes in other disease areas. For example, if we used all admissions as denominators, an unrelated increase e.g. in admissions for injuries (S00-99) would change the relative frequency of admissions for cardiovascular disease. This is not the case if only the frequency of cardiovascular disease relative to each other is analysed.

Comment #1.6: Have the authors validated their EHRtemporalVariability R package they developed? For instance, was it used in previous pilot studies to test it?

Author response:

The methods implemented in the EHRtemporalVariability have been used and validated in the following publications:

[1]: Sáez C, Rodrigues PP, Gama J, Robles M, García-Gómez JM. Probabilistic change detection and visualization methods for the assessment of temporal stability in biomedical data quality. Data Mining

and Knowledge Discovery. 2015;29:950–75. <https://doi.org/10.1007/s10618-014-0378-6>

[2]: Sáez C, Zurriaga O, Pérez-Panadés J, Melchor I, Robles M, García-Gómez JM. Applying probabilistic temporal and multisite data quality control methods to a public health mortality registry in Spain: A systematic approach to quality control of repositories. *Journal of the American Medical Informatics Association*. 2016;23:1085–95. <https://doi.org/10.1093/jamia/ocw010>

[3]: Sáez C, García-Gómez JM. Kinematics of Big Biomedical Data to characterize temporal variability and seasonality of data repositories: Functional Data Analysis of data temporal evolution over non-parametric statistical manifolds. *International Journal of Medical Informatics*. 2018;119:109–24. <https://doi.org/10.1016/j.ijmedinf.2018.09.015>

[4]: Perez-Benito, F.J., Saez, C., Conejero, J.A., Tortajada, S., Valdivieso, B. and Garcia-Gomez, J.M., 2019. Temporal variability analysis reveals biases in electronic health records due to hospital process reengineering interventions over seven years. *PLoS one*, 14(8).

The R package itself is open source and is published in the official, validated CRAN repository. It has been downloaded more than 3,500 times since March 2019.

Comment #1.7: Table 1 provides useful information on possible causes of observed coding variability in CPRD GOLD and HES datasets. It would be useful if the authors would give more specific steps to researchers using electronic health records to follow in future studies to take the reported variability on clinical coding into account.

Author response:

It is difficult to give generic advice on how to deal with the observed coding variability. In general, the first step in assessing the impact of temporal variability is to investigate any changes in effect size over time. Simple methods to do so in regression analysis involve stratified/separate analysis (for discrete, abrupt coding changes) and continuous interaction effects (for gradual changes). Changes linked to subgroups (e.g. the observed change in socio-economic status due to drop out of practices) might be accounted for via mixed model analysis. We added a fourth column “Possible solutions” to Table 1 sign-posting some of these possibilities.

Comment #1.8: It is understandable that the IGT plots contain many information and the authors have presented two formats (with and without age) and added arrows to improve visualisation. But still it is hard to spot changes at individual months of interest. If practical, I suggest considering to include a plot or two either: over a shorter period of time that showed more clinically relevant changes on coding variability (e.g. before and after QOF launch); or to plot using yearly than monthly points to improve plot clarity.

Author response:

We thank the reviewer for pointing out the remaining difficulty in dissecting the IGT plots. To try and alleviate the issue, we have created detailed subplots corresponding to the highlighted periods a, b, c and d in both IGT plots. In the interest of keeping the manuscript succinct, these additional plots were added to the supplement and a reference to them was added to the captions of Figure 2 and 3.

Comment #1.9: The authors stated a limitation that the CV conditions chosen for inclusion in this study represented a convenience sample from a previous project and hence no other clinical conditions were examined. Based on that, what other conditions would the authors recommend to be examined for temporal code changes in future research?

Author response:

Important targets for future research are commonly researched conditions and confounders. A prime target from our point of view are those chronic conditions included in the Quality and Outcomes Framework (<https://digital.nhs.uk/data-and-information/data-tools-and-services/data-services/general-practice-data-hub/quality-outcomes-framework-qof>). Some of the authors are currently investigating changes in the coding of all ICD-10 categories in HES (see discussion lines 329-332), and of hypertension, chronic kidney disease and type II diabetes in particular.

However, we also take the view that dedicated, published validation can only be a partial solution. The limitations of such an approach become immediately apparent for example in CPRD. For our results presented in this study, we analysed the CPRD GOLD database, which is based on the practice management software Vision. CPRD recently introduced CPRD Aurum, a new database that gathers data from practices with EMIS software. Important differences might exist between the software systems that we haven't been able to investigate here, as our data is based solely on CPRD GOLD. Validation studies can only ever provide a snapshot, and future studies relying solely on results presented in this study might be ignorant of future changes. Researchers must be aware that the use of slightly different data, study periods or code lists may all introduce new temporal trends.

Instead of providing a definite quality assessment of cardiovascular disease recording in CPRD and HES, a central aim of this paper is to provide researchers with simple and powerful tools to examine temporal code changes specific to their data. By making the assessment for temporal changes an integral part of their analysis workflow, researchers can report any irregularities relevant to their research question. As a result, researchers as well as readers can be more confident that findings reflect true underlying effects rather than temporal artefacts.

Comments from reviewer #2

Comment #2.1: Before commenting on the analysis, I would seriously question the number of 451 English hospital trusts found in 2014/15. There were around 180 acute trusts in 2000, now down to about 135 acute non-specialist trusts in 2018, 17 acute specialist trusts and 54 mental health trusts. Even accounting for mergers and the inclusion of mental health trusts, it seems hard to see where the figure of 451 trusts comes from. This would cast doubt on the whole analysis.

Author response:

We thank the reviewer for pointing out this important inconsistency. The term "trust" as well as the number of 451 participating institutions was quoted directly from the Hospital Episode Statistics (HES) Data Resource Profile (<https://www.ncbi.nlm.nih.gov/pmc/articles/PMC5837677/>). However, we agree that this is inconsistent with how the term "trust" is used elsewhere in the NHS and could indeed cast doubt on the validity of the analysis.

We changed "hospital trusts" to "hospital providers" in lines 38 and 146 of the revised manuscript. This terminology is in line with both the HES Data Resource Profile and Health & Social Care Information Centre's (HSCIC) HES APC summary report from 2014/15 (<https://www.google.com/url?q=https://web.archive.nationalarchives.gov.uk/20180328135541/http://digital.nhs.uk/media/27811/Hospital-Episode-Statistics-Admitted-Patient-Care-England-2014-15-Summary-Report/Any/hosp-epis-stat-admi-summ-rep-2014-15-rep&sa=D&ust=1577980908732000&usg=AFQjCNEP2IIClpOL7kcud7vSYbTYtFPUIg>)

Comment #2.2: The authors state that variables extracted from CPRD GOLD showed a gradual trend from one month to the next between 2001 and 2007. They suggest a monthly increase in the

prevalence of a single disease from 1% to 2.2% or from 50% to 55%. I may be misunderstanding their statement, but a monthly increase of 55% would seem to be impossible. I suspect they mean over the whole period.

Author response:

We apologise for the confusion. We try to expand on the statement here and have made changes to the manuscript to clarify our point (lines 232-236).

We are aware that changes in Jensen-Shannon distance (JSD)/month are hard to intuit. In order to give the reader a more intuitive measure of how to interpret a change in JSD of $\sim 1.5 \times 10^{-3}$ per month, we compare it to what the same JSD/month would mean if we only analysed changes in a single variable. In this case, a JSD/month of $\sim 1.5 \times 10^{-3}$ would be observed if the prevalence of the variable in our population changes from 1% in one month to 2.2% in the next. Thus, the sum of changes in all 7 variables changes the joint distribution as much as if the prevalence of one binary variable changes from 1% to 2.2%

Furthermore, the JSD depends on the base prevalence. A change from 1% prevalence to 2.2% (uncommon trait; 120% increase) is a relatively more severe change than a change from 51% to 52.2% (common trait; 2.3% increase). As a result, a larger absolute change (e.g. from 50% to 55% or 51% to 57%) is needed for common traits to result in the same JSD.

Comment #2.3: What the authors show is a change in coding, particularly within HES from vague codes to more specific codes, suggesting improved coding and probably better diagnostics (with an increase in CT scans for stroke and troponin trusts in suspected AMI). Within CPRD coding appears to have been driven by QOF payments. None of this is surprising or new findings. Much earlier research has suggested an improvement in coding over time (E.M. Burns, E. Rigby, R. Mamidanna, A. Bottle, P. Aylin, P. Ziprin, O.D. Faiz. Systematic review of discharge coding accuracy. Journal of Public Health, Volume 34, Issue 1, March 2012).

Author response:

We thank the reviewer for the additional input on reasons for improved coding and the additional reference. We added the suggestions to the discussion section of the manuscript (lines 306-307 and 318-319)

We further agree that by itself, a finding of improved coding after the introduction of aren't necessarily new. Instead, the fact that we were able to repeat those findings attests to the validity of the methods used in our study. Without any prior specification, our methods were able to pick up previously reported changes in coding in the early 2000's for both CPRD and HES. This highlights how the methodological framework employed by the authors can be used by future researchers to reliably detect unreported changes present in their own data.

Besides the use of a new and flexible framework, novel findings of specific changes in CPRD and HES are:

- a dedicated and up-to-date analysis of coding changes in cardiovascular disease, which to our knowledge has not been performed before
- a demonstration of the relative magnitude of changes in socio-economic composition caused by the recent drop in practices contributing to CPRD GOLD
- the identification of very distinct, biyearly changes cardiovascular disease coding in HES

Comment #2.4: Although these national trends are not new findings, the methods suggested might be useful in examining temporal trends in individual trust coding practices, and therefore could be employed to help interpret trust level indicators dependent on diagnostic coding. However, I am struggling to see how the authors' suggested methods are any better than simple time series analyses, but am happy to be persuaded otherwise. I note that none of the authors are clinicians, and the paper might benefit from clinical input to help explain some of the changes,

Author response:

Time series analyses are a valuable instrument to analyse changes in numerical data. For example, they allow to closely monitor the evolution of a single code's incidence over time. However, they struggle to monitor multitype or multimodal data, such as the evolution of many codes in our HES sub-analysis. A large number of individual analyses would have to be performed, one for each code. Analysis is further complicated if additional control variables (age, gender, socio-economic status) must be accounted for and/or are of interest in their own right.

Temporal variability metrics can readily monitor and explore numerical, categorical and multivariate data. For example, researchers can monitor the evolution of all codes simultaneously, and also combine them with other variables such as age. In other words, these methods allow researchers and data curators to monitor joint distributions rather than only univariate or conditional distributions. These types of analysis are not straightforward with traditional time series analysis.

Additionally, time series analyses are not suited to detect temporal subgroups. Instead, they only track changes on a statistic of interest. Examples of temporal subgroups in this study are the biyearly time batches identified in HES after 2010.

An in-depth discussion of the advantages of variability metrics over more traditional analysis methods can be found in: Sáez C, Robles M, García-Gómez JM. Stability metrics for multi-source biomedical data based on simplicial projections from probability distribution distances. *Stat Methods Med Res* 2017;26:312–36.

Finally, we fully agree with the reviewer that clinical input is invaluable in interpreting the findings. Clinical input was provided one of our co-authors, Dr Rob Aldridge, who is a consultant in public health. We further presented and discussed our conclusions with Dr Anoop Shah, a consultant in general medicine with extensive experience in working with EHR records. We extended our description of Dr Shah's help in the acknowledgment section to clarify the input provided by Dr Shah.

VERSION 2 – REVIEW

REVIEWER	Dr Salwa Zghebi University of Manchester, UK
REVIEW RETURNED	14-Jan-2020
GENERAL COMMENTS	The authors have addressed my comments satisfactorily and I have no further comments. Thank you.